# Building Water Quality Commissioning in Healthcare Settings: Reducing *Legionella* and Water Contaminants Utilizing a Construction Scheduling Method

**Molly M. Scanlon** [1,*] , **James L. Gordon** [2] **and Kelly A. Reynolds** [1]

1 Department of Community, Environment and Policy, Mel and Enid Zuckerman College of Public Health, University of Arizona, Tucson, AZ 85724, USA; reynolds@arizona.edu
2 Gordon Consulting, Coronado, CA 92118, USA; jimgordonarchitect@yahoo.com
* Correspondence: mscanlon@arizona.edu

**Abstract:** Construction activities in healthcare settings potentially expose building occupants to life-threatening waterborne pathogens, including *Legionella*. The lack of a building water quality commissioning (BWQC) process has been identified as a substantial construction risk factor associated with disease cases and deaths. A BWQC schedule method was developed as a technical note to address gaps between the construction, commissioning, and operation phases of work to establish water quality and safety for a building water distribution system. The BWQC schedule method enables healthcare organizations to meet commissioning criteria set forth in guidelines and regulatory requirements for implementing a water management program (WMP) prior to initiating patient care operations. The authors used Office Timeline® Pro+Edition V7.02, Office Timeline LLC, Bellevue WA 98004, USA to depict a Gantt chart as a BWQC schedule listing key project tasks and milestones of construction and water management activities. Design and construction professionals, in conjunction with healthcare organizations, should examine the BWQC construction schedule method and customize it for project-specific implementation. Additionally, building owners should consider incorporating the method into an organization's construction policies for a standardized approach to BWQC practices.

**Keywords:** construction; commissioning; healthcare facilities; *Legionella*; premise plumbing; project management; scheduling; water management; water quality; water safety plan





## 1. Introduction

The lack of a building water quality commissioning (BWQC) practice for a building water distribution system (BWDS) has contributed to disease cases and deaths from *Legionella* and other waterborne pathogens [1,2]. Public health agencies and healthcare organizations have reported case studies (*n* = 31) resulting in disease cases (*n* = 894) and deaths (*n* = 112) from waterborne pathogens associated with poorly executed construction and commissioning activities [2]. Outbreaks in healthcare settings continue to occur related to construction and commissioning activities and often go unreported [3]. These events are categorized as healthcare-associated infections, in which a patient acquires an infection while staying in a healthcare setting and undergoing observation or treatment [4,5]. Hospital patients or skilled nursing residents who have underlying diseases or immunocompromised systems (e.g., diabetes, chronic lung conditions, cancer, heart disease, weakened immune systems, burns, or transplant recipients) are more vulnerable to waterborne pathogen disease than the general population [5,6]. Any device with a water reservoir connected to or utilizing the potable water system can become contaminated (e.g., bathtubs, showerheads, sinks, drains/traps, dialysis water supply, medical equipment, ice and ice machines, or premise plumbing system components) [5]. Patient exposure primarily occurs through ingestion with aspiration or aerosol inhalation.

The building construction and shut-down processes are known to create low or no-flow conditions, resulting in high water age throughout the course of construction activities [7–9]. High water age (i.e., water sitting in premise plumbing and becoming stagnant) in the BWDS increases the potential for the growth and spread of waterborne pathogens [10,11]. Since 2017, United States (US) hospitals, skilled nursing facilities, and critical access hospitals have been required to develop a water management program (WMP) prior to the operations of a facility to receive US Federal reimbursement for patient care services [12]. In the US, a WMP is implemented in alignment with the American National Standards Institute (ANSI)/American Society of Heating, Refrigerating, and Air-conditioning Engineers (ASHRAE) Standard 188, Legionellosis: Risk Management for Building Water Systems [7], as well as the recommendations provided by the US Centers for Disease Control and Prevention (CDC) Toolkit: Developing a Water Management Program to Reduce *Legionella* Growth and Spread in Buildings [13]. ANSI/ASHRAE Standard 188 [7] requires hazardous conditions to be addressed during various phases of BWDS design, construction, and commissioning to assure water quality and safety. Other authorities having jurisdiction (AHJ) over US healthcare operations for design, construction, or financial reimbursement have subsequently stated similar WMP requirements, inclusive of construction and commissioning phases of work [14–16]. Additionally, international countries require healthcare facilities to have a similar process (i.e., a water safety plan) in place [1,3,17–20] to meet their specific country's public health policies to control *Legionella* and other waterborne pathogens such as nontuberculosis mycobacteria (NTM), *Pseudomonas*, or others.

Healthcare WMP teams have been given minimal methods, tools, or training to coordinate this risk management process within the overall construction process [21]. WMP standards state the minimal potable water activation approach as disinfection and flushing of the premise plumbing before beneficial occupancy [6,7]. Limited guidance on additional flushing or disinfection is given if beneficial occupancy is delayed. Further, infection prevention and control practitioners, who are often members of a healthcare WMP team, are required to perform an infection control risk assessment (ICRA) for airborne and waterborne pathogen control during construction activities [21,22]. However, the emphasis has been primarily on airborne pathogen controls (e.g., *Aspergillus*), with little emphasis on waterborne pathogens [23,24]. Figure 1 depicts a broad range of construction activities undertaken in healthcare facilities and the relationship between project types and BWDS disruptions impacting water age [21]. Additionally, architectural, engineering, and construction (A/E/C) professionals and commissioning agents (CxA) have minimal knowledge about water science, water contaminants, WMP requirements, or analytical laboratory water analysis [25,26]. These professionals need core competency in water safety skills to effectively communicate and perform services with the building owner and their WMP safety teams [1].

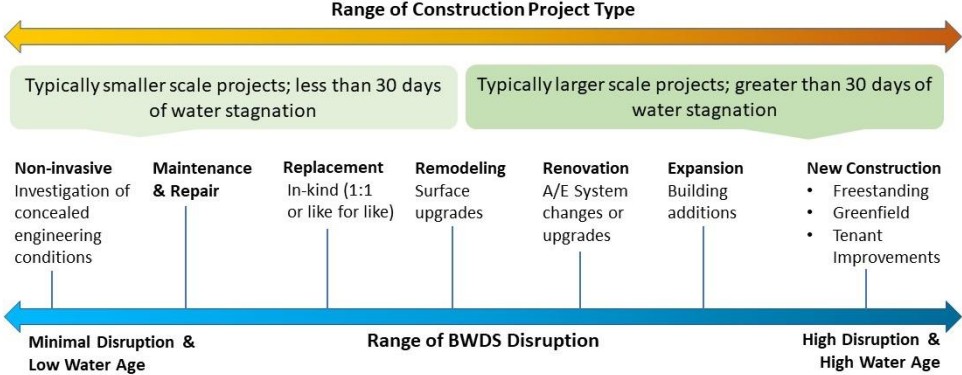

**Figure 1.** Range of healthcare facility construction project types and BWDS disruption.

Compounding the situation, A/E/C professionals use building rating and certification systems to focus on green building design and water efficiency (i.e., energy and water

conservation), not water quality and safety [11,27–29]. Water efficiency creates high water-age conditions in direct conflict with known water science. Balancing water efficiency with water quality and safety objectives is essential to reducing the likelihood of healthcare-associated infections from BWDS growth and spread of waterborne pathogens. To obtain this balance, A/E/C professionals need to integrate an effective and compliant BWQC process into the overall scope of construction work [1,25]. Simply telling construction teams to flush and disinfect the BWDS does not adequately address the wide range of construction types, the level of BWDS water disruption, and the known construction risk factors to reduce unintended consequences [1,2,21,25].

### 1.1. Healthcare Construction, Scheduling, and Commissioning

Healthcare construction projects are considered one of the most complex building types to assemble due to the nature of construction, engineering systems, and the patient safety regulatory environment [30,31]. In 2016, a US hospital construction survey identified 1300+ healthcare construction projects with a financial investment of USD 97 billion [32]. Hospitals are under a cycle of continuous physical facility improvements due to aging infrastructure, increased patient loads, and innovative medical treatments and technologies. To coordinate building systems for patient safety, complex project management and scheduling are considered essential methods of healthcare construction [31,33]. A/E/C professionals commonly use construction scheduling methods to help manage and organize key project tasks and milestones, labor and resources, product delivery and installation, and final tests and inspections for a timely and cost-effective project delivery process [34]. The simple and easiest communication tool for initial project scheduling is the Gantt chart (i.e., bar chart) [33,35]. A/E/C and CxA professionals routinely use Gantt charts, which are still preferred for project planning and scheduling purposes. Healthcare building owners depend upon project planning and scheduling activities to determine the overall project duration, the impact on patient care operations, planning for service utility disruptions, including potable water and heating systems, and meeting regulatory requirements [33].

Commissioning for any building system is the practice of verifying, documenting, and optimizing a building system to meet a pre-defined set of requirements and objectives from the codes and standards and the building owner's criteria [18,32,36]. The commissioning plan and process are generally implemented by either A/E/C or CxA professionals who are hired to optimize energy efficiency and building system performance. Commissioning for building system performance standards focuses on verifying if the building system is working as designed. The current water commissioning process lacks accountability [3,20,26] and does not routinely validate if the building system is working to minimize water contaminants through the premise plumbing [37,38]. The general assumption has been that new building systems would not contain contaminants [39]. Water is an organic substance with chemical properties that degrade over a short period of time [11,27]. Even in new building piping, water is capable of growing and spreading bacteria and waterborne pathogens such as *Legionella* [40], *Pseudomonas* [41], or NTM [42]. To our knowledge, no construction scheduling method or task sequencing has been developed to demonstrate how the A/E/C and CxA industries can respond to water quality and safety in healthcare settings.

### 1.2. Current Construction Industry Practices

Current industry practice allows the construction team to assemble BWDS with minimal oversight (see Figure 2). After the contractor fills the BWDS, the water in the premise plumbing often sits for months [40] or even years [43] before the facility is occupied for public use. Typically, the contractor will minimally utilize the BWDS for mixing water with aggregates and mortars, grouts, or maybe for temporary restrooms for the construction staff [44]. Otherwise, the water within the premise plumbing remains dormant for extended periods of time while the remaining building systems are assembled. The general contractor and plumbing subtrades check the BWDS for limited performance criteria such as water

pipe leakage, plumbing fixtures dispense water, adequate water pressure, and the water heating system will circulate and register a water temperature appropriate for building occupant usage [44,45]. If water quality and safety are considered, the tasks typically involve BWDS disinfection toward the end of the construction project, paired with testing for *E. coli* and coliforms related to potable drinking water [38]. Other key water parameters are not measured, and other waterborne pathogens associated with clinical disease are not routinely tested. Therefore, the building owner has no assurance that the water quality is safe for patient care hygiene (e.g., sinks and showers) or treatment (e.g., medical equipment, instrument sterilization, or ice machines). Yet, healthcare providers are required to have a WMP risk management plan for water quality and safety ready for patient care operations prior to building occupancy [7,12].

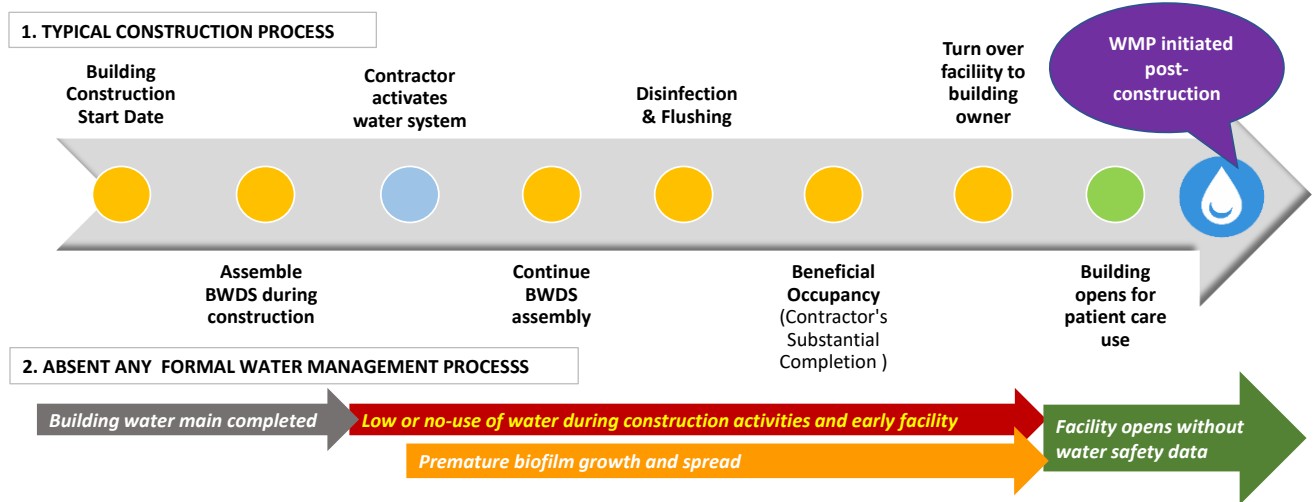

**Figure 2.** Current healthcare building construction practice without water management.

The purpose of this technical note was to demonstrate a construction scheduling method to coordinate BWQC activities during periods of construction to reduce the risk of illness, injury, or death from waterborne pathogens and other water contaminants. The novelty of this technical note is our use of a common construction project scheduling method, the Gantt chart, to easily integrate WMP with BWQC practices into the normal sequence of building construction without violating existing water science evidence-based practices, water management public policy, or regulatory guidance documents. This scheduling tool is intended to address the gap between BWQC requirements and WMPs used for ongoing operations in healthcare settings.

## 2. Materials and Methods

### 2.1. Construction Scheduling Method

Gantt charts (i.e., bar charts) are used as a construction scheduling method to graphically represent project information for management, communication, and completion of construction tasks over a designated time period [35]. When constructing a Gantt chart, the project must be broken down into incremental steps and grouped into homogeneous categories (e.g., construction vs. water management) [35]. Within each category, tasks and sub-tasks are developed, including key milestones. It is important not to overburden the schedule with too many task/activity line items to allow for basic communication among interdisciplinary team members [35].

For this technical note, the authors used Office Timeline® Pro+Edition V7.02 by Office Timeline LLC, Bellevue WA 98004 USA, a desktop software program [46], to develop exemplar construction schedules to depict a list of tasks (y-axis) over a duration of time (x-axis) [35]. Office Timeline® operated as a toolbar add-in with pull-down menus and templates within Microsoft Power Point® Microsoft Office Home and Business Edition V2019

by Microsoft Corporation, Redmond WA 98052 USA, for seamless integration to allow for figures to be created that are simple, readable, and organized for journal publication. Office Timeline® was exported as digital file formats (e.g., graphic filename.png). Other project or construction management software programs could equally be used that contain a feature to develop or import Gantt charts. Gantt chart information was initially broken down by discipline (i.e., the overall project, construction, and water management) and subsequently by phase (i.e., building shell and core, tenant improvements).

## 2.2. BWQC Construction Schedule Tasks and Milestones

Key milestones within a project schedule were depicted with the intent of coordinating interim progress toward the overall final goal of opening the building with safe water. For the purpose of this technical note, the authors have depicted a limited number of typical BWDS project tasks and milestones (see Table 1) representative of typical new building construction over approximately a 12-month period. For a comprehensive list of construction tasks, challenges, and hazardous conditions for assembling a BWQC project schedule, see Supplement S1: Building Water Quality Commissioning Project Schedule Checklist. The BWQC project scheduling method was developed to align with the practice of the *Legionella* risk management process established in ANSI/ASHRAE Standard 188 [7] and its companion ANSI/ASHRAE Standard 514 [6,47] for other water-related physical, chemical, or biological hazards.

## 2.3. BWQC Roles and Responsibilities

Determining members of a WMP team and a designated team leader is the responsibility of the building owner's senior organizational leadership [6,7]. The team leader is responsible for organizational authority and developing and implementing the WMP, inclusive of construction and commissioning activities. For the purposes of establishing the BWQC project schedule efforts, we will assign the title of building owner representative (BOR). An interdisciplinary healthcare WMP team is required to have members familiar with facilities management (FM), infection prevention and control (IPC), clinical leadership (CL) for patient care operations, and occupational and environmental safety (OES) for staff and patients [6,7]. One or more members must be familiar with the BWDS design and any potentially hazardous conditions (e.g., construction and commissioning activities) that may impact water management related to physical, chemical, and microbial hazards [6].

Specific roles and responsibilities for the BWQC task list are also to be established and implemented by the WMP team [6,8]. The suggested leader(s) for each task are listed in Table 1 as a preliminary starting point. The BWQC project schedule would likely be implemented based on the specific knowledge, skills, and abilities of the designated team members. For example, construction-like skills (e.g., creating a project schedule, conducting the pre-construction meeting, and assembling the BWDS) would likely fall to the construction team members. A CxA specializing in water system performance criteria could be hired by either the general contractor or the building owner to ensure the BWDS will be operating as designed. The building owner's roles and responsibilities would include overall project coordination, patient care safety, and determining acceptance of water quality. Additionally, the building owner may hire a water management specialist (WMS) as a consultant specializing in WMP development, analytical laboratory testing, and interpretation of water test results. AHJ may recommend independent water quality testing (i.e., not a subtrade to the general contractor) to ensure the BWDS meets a minimum standard of care prior to opening for patient care operations [19].

**Table 1.** Construction Scheduling: Key Project Tasks and Milestones. Abbreviations for Key Team Leader(s) roles and responsibilities: AE = architect/engineer professional; AHJ: authority having jurisdiction; BOR = building owner representative; CL = clinical leadership; CxA = commissioning agent; FM = facility manager; GC = general contractor; IPC = infection prevention and control; N/A= not applicable; OES = occupational/environmental and safety specialist; SUB = subtrade or subcontractor; and WMS = water management specialist/laboratory consultant.

| Discipline | Phase | Key Tasks/ Milestones | Key Team Leader(s) | Description |
|---|---|---|---|---|
| Overall Project | N/A | Project Kick-off | GC, BOR | The first day the GC is contractually engaged to perform work on the construction project [44]. |
| | | Water Activation | GC, CxA, BOR | The first date water is flowing in any section or component of BWDS beyond the building's main point of entry [3,44]. |
| | | Beneficial Occupancy | GC, CxA, BOR, FM | The date GC has substantially completed the construction project and the building owner's staff can legally and safely occupy the building. It is also referred to as the date of substantial completion for fire and life safety conditions [6,7]. |
| | | Water Quality Approval | BOR, CL, IPC, OES, WMS, CxA, AHJ | The building owner formally accepts the water quality results from the GC and deems the water quality appropriate for patient care operations [6]. |
| | | First Day of Patient Admissions | BOR, FM, IPC, CL, AHJ | A date on which the building owner plans to begin legal healthcare operations after approvals from local, state, or national AHJ and admit patients to the healthcare facility [44]. |
| Construction | Building Shell and Core | Pre-Construction Review Meeting | GC, SUB, A/E, BOR, FM, WMS | Similar to other core building systems, conduct a review of the BWDS design documents to ensure the system and its components meet the building owner's WMP performance criteria (e.g., incoming water, hot and cold water distribution systems, hot water storage, and fixture types). Next, coordinate a specific meeting for the WMP/BWQC safety team to review project goals and objectives, using the overall construction project schedule as a document for discussion and agreement [44]. |
| | | Assemble A/E core building systems for plumbing | GC, SUB | Divide this task into levels, floors, or key departmental sections to install the premise plumbing core components of the BWDS main, branches, and risers [44]. |
| | | Building Water Flow | GC, SUB, CxA | The amount of water passing through a pipe at any given time is water flow. Water flow is affected by the width of a supply pipe. Fixture water flow can be altered by aerators, lamer flow screens, or auto-fixtures [9]. |
| | Tenant Improvements | Assemble interior A/E systems for plumbing | GC, SUB | Divide this work effort into departments and rooms on each floor and install the premise plumbing interior finish components of the BWDS, including local branching, valves, plumbing fixtures, and fixture trim, as well as pressurizing the system [44]. |
| | | Disinfection/ Purging | GC, SUB, CxA, BOR, FM, WMS | Disinfection: A form of hyperchlorination of the BWDS (chemical disinfection) toward the later stages of building construction. Chemical disinfection injects high quantities of chemicals into the BWDS above allowable drinking water limits. This water is purged eventually from the system [6–8]. Purging: A form of elimination of water from the BWDS in large quantities to turn water over rapidly after water disinfection activities. This is a distinct activity separate from periodic fixture flushing. |
| | | GC Substantial Completion | GC, SUB, CxA | The GC has completed a substantial portion of the project (e.g., 90% complete), and 100% of all building fire and life safety issues (e.g., fire exiting, stairs, fire alarms, smoke dampers, etc.) are complete for people to occupy the structure safely while other work (e.g., punch list items) can be completed. |
| | | Building Turnover/ Owner Move-in | BOR, FM, CxA, AHJ | The period of time in which the building owner's facility engineering and maintenance staff take possession of the building to initiate preliminary facility operations, transition planning, and move-in/setup. The building owner is preparing for review with AHJs (national, state, or third-party agencies) for approval to admit patients into a licensed healthcare setting. |
| | | Facility Operations | BOR, FM | The building owner's staff takes 100% control of the building and its engineering systems, including BWDS water quality and safety. The GC has completed the project contract. |

**Table 1.** *Cont.*

| Discipline | Phase | Key Tasks/ Milestones | Key Team Leader(s) | Description |
|---|---|---|---|---|
| Water Management | WMP/BWQC Team and Documents | WMP/BWQC Review Meeting | GC, CxA | Coordinate a meeting for the construction/commissioning water safety team to review WMP/BWQC project goals and objectives using the overall project construction schedule as a document for review, discussion, and agreement [44]. |
| | | WMP/BWQC Risk Assessment | BOR, CL, IPC, OES, FM, WMS | The water safety team should review the scope of the project and determine the risk of the project to building occupants (existing occupants and future occupants) [6,7,21]. |
| | | Developing WMP/BWQC Plan | BOR, FM, IPC, OES, WMS, GC | Create a WMP/BWQC document with the designated safety team for BWQC implementation during construction and commissioning activities per ANSI/ASHRAE 188 and 514 [6,7]. |
| | | Transition BWQC to WMP Documentation | GC, SUB, CxA, BOR, FM, WMS | At the conclusion of a construction project, the BWQC documentation is handed over to the building owner. The building owner transitions the BWQC from the safety team assigned during the construction/commissioning project phase toward a team focused on performing ongoing facility operations. Formal WMP team meetings are held to adjust the documentation accordingly with changes to hazard control options for ongoing facility operations [6]. |
| | BWQC Hazard Controls | Flushing Activities | GC, SUB or BOR, FM, WMS | A hazard control to be implemented periodically to reduce the high water age and increase the movement of water through the BWDS. Establish a flushing protocol to occur on $x$ fixtures for $y$ day(s) per week and $z$ minutes per fixture valve(s) (hot and cold) [8–10,48]. |
| | | Disinfectant Residual Readings | GC, SUB or BOR, FM, WMS | Monitoring designated fixture locations for residual disinfectant measurements to ensure a minimum level of disinfectant residual resides within piping distribution and fixtures after the water is processed through the BWDS [10]. |
| | | Temperature Checks | GC, SUB or BOR, FM, WMS | Monitoring the temperature of water (hot and cold) at designated fixtures to ensure water temperature remains within range as defined by the water management program [10,19]. |
| | BWQC Confirmation | BWQC Verification | BOR, WMS, CxA | The building owner and their designee(s) will verify that hazard controls are being implemented as designed for the WMP/BWQC plan on a routine basis (e.g., monthly meetings) by having periodic check-ins, including the GC [6,7]. |
| | | BWQC Validation Testing | BOR, FM, CL, IPC, WMS, CxA | WMP/BWQC team to determine the type, quantity, location, and method of analytical testing for microbials (e.g., *Legionella*, *Pseudomonas*, NTM) and/or chemicals (e.g., lead, copper) [10] at each sampling date and time (T0, T1, and T2) [6,8,19,39]. |
| | | T0 Baseline | WMS, CxA, BOR, FM | Time Zero—baseline analytical testing; determine system performance after hazard controls (e.g., flushing, monitoring disinfectant residual, temperature) are implemented and prior to chemical disinfection [19,21]. |
| | | T1 Microbial + Chemistry | WMS, CxA, BOR, FM | Time One—analytical testing conducted after chemical disinfection and purging of BWDS; performed post disinfection after the system is at rest for at least 48 h [19], and suggest sampling is undertaken by an accredited laboratory independent of the GC [19] (lab processing times will vary based on analytical test types being performed). |
| | | T2 Retests and Follow-up | WMS, CxA, BOR, FM | Time Two—analytical testing after T1 is successfully completed. Perform T2 between the completion of T1 and the first day of patient care operations for any retests or additional testing to ensure the BWDS has maintained a low water age after the chemical disinfection was performed. |

## 3. Results

Utilizing the methods outlined above, the authors developed a conceptual BWQC schedule inclusive of when to initially prepare and transition WMP documents. Rather than waiting until the building is open, occupied, and operational to start the WMP process (refer back to Figure 2), this method would begin at the start of the building construction process (see Figure 3). The construction water safety team would utilize the same ANSI/ASHRAE Standard 188 and Standard 514 water risk management processes [6,7]. The BWQC hazard control process would be implemented immediately when the contractor activates the BWDS for water flow in any section of the premise plumbing piping.

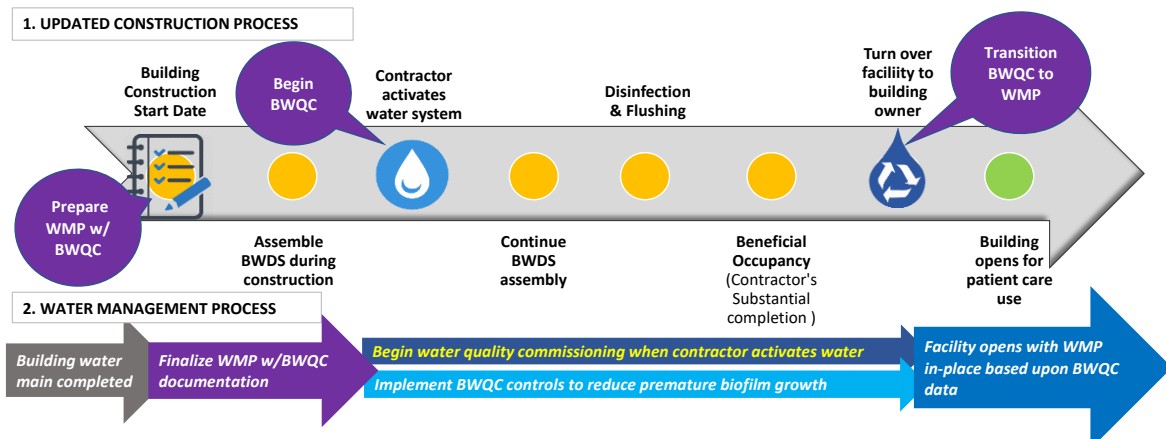

**Figure 3.** Proposed healthcare building construction process with water management and water quality commissioning practice. BWDS = building water distribution system; BWQC = building water quality commissioning; and WMP = water management program.

Hazard controls would be more robust during the construction and commissioning phases to reduce stagnant water conditions due to the building being unoccupied. At the end of the project, when construction and BWQC efforts are completed, the building owner will transition the BWQC plan toward a WMP for ongoing building operations. Using this process, the facility opens with known water quality and safety data that have been observed and recorded over several months. This process acts as confirmation that the BWDS and WMP, under proper team supervision, can successfully maintain water quality and safety parameters, as well as continue to reduce the risk of disease cases emerging once the patient population is admitted to the facility.

### 3.1. Gantt Chart Results

After the conceptual project construction approach and water management process are understood, the GC will prepare a more detailed BWQC construction project schedule with input from mechanical and plumbing subtrades, design architects and engineers, CxA agents, the building owner, and any infection prevention or safety officers involved in the project. Using the Gantt chart method and the list of key tasks and milestones outlined in Table 1, we illustrated two Gantt chart options for establishing a BWQC construction project schedule for a new hospital building.

Gantt chart schedule A (see Figure 4) assumed the GC would need access to potable water early in the overall project construction process for temporary restrooms or for construction hydration processes (i.e., mixing mortars or types of cement). Upon water activation (Figure 4 as of April 1), the WMP/BWQC team and documentation must be in place to begin the hazard control process. BWQC efforts continue throughout the project construction process to monitor water quality and safety until the end of the project. Prior to BWDS disinfection, a sequence of validation testing (i.e., analytical laboratory testing) begins to determine how the BWDS will perform over time (i.e., T0, T1, and T2). The WMP/BWQC team must allow for adequate time in the project schedule for all analytical laboratory testing, including sample collection, shipping, lab processing, and reporting of results, needs to be considered. These laboratory timeframes can range from 5 to 30 business days, depending on the analytical test method being performed. If the validation test results are negative (i.e., no results were above the minimum recommended detection limits), the water quality results would be presented to the building owner's WMP/BWQC team for acceptance and approval (Figure 4 as of December 15). If there were elevated (i.e., positive) results, the WMP/BWQC team must review the data and determine if these results are systemic or localized to a specific fixture. Widespread elevated detections would suggest a systemic problem and a need for a further assessment of the overall BWDS

commissioning efforts. Elevated results may lead to schedule delays depending on the timing of receiving results in the context of the overall construction project. After the BWQC validation response is implemented (i.e., hazard controls specifically implemented to address the elevated results), T2 validation testing would either be a retest for systemic challenges or follow-up testing to address isolated fixture conditions.

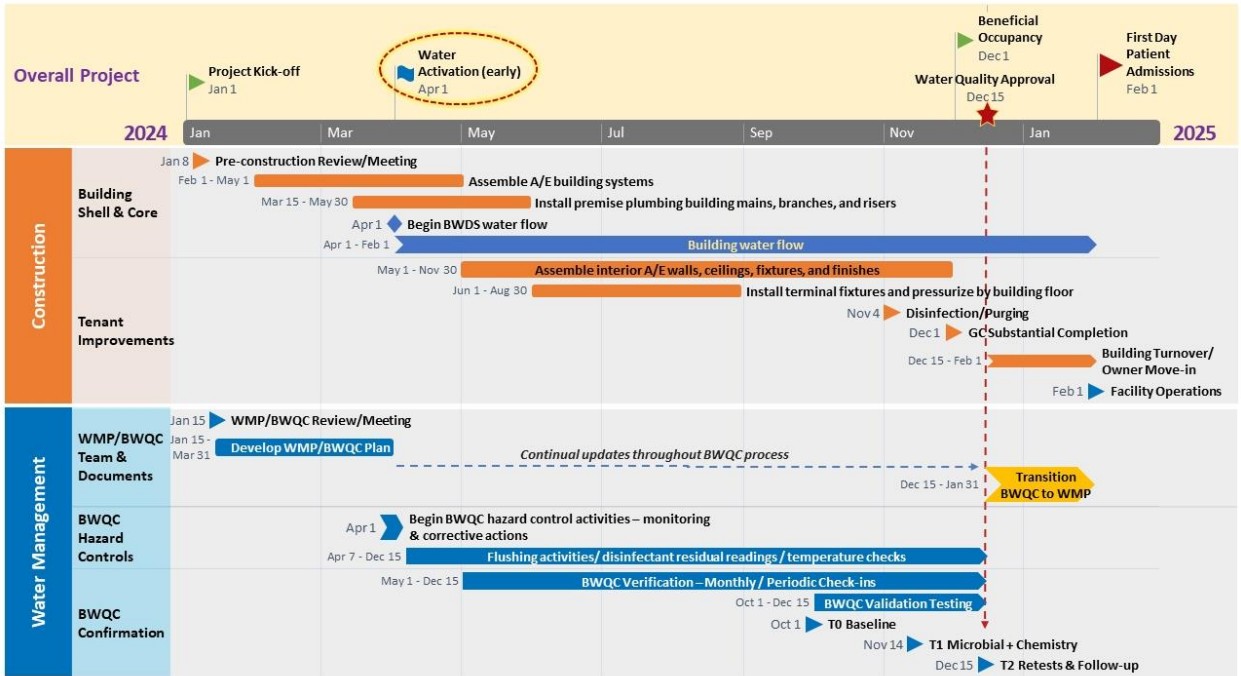

**Figure 4.** Gantt Schedule A—BWDS water activation occurring early in the building shell and core construction phase.

Gantt chart schedule B (see Figure 5) assumed the general contractor delayed BWDS potable water start-up and used an alternative source for accessing construction water (Figure 5 on April 1) [44]. Contractors could consider using temporary piping runs to designated locations within the construction zone. This approach would not utilize the building owner's final premise plumbing. Rather, the temporary piping would be used for construction activity water access and terminated (i.e., abandoned and removed) later in the project. This allows water activation to be delayed as late as possible in the same project schedule of events [3]. Gantt chart schedule B illustrates a 4-month delay in water activation (Figure 5 on August 1). The benefit of delaying water activation was to avoid high water stagnation, diminish premature biofilm growth and spread, and reduce the labor involved in BWQC activities. This approach would also likely reduce water consumption and increase water efficiency for BWQC activities. All other construction dates within Gantt chart schedule B essentially remain the same, with water management dates compressed into shorter durations.

After beneficial occupancy using either Gantt chart schedule A or B, the BWQC efforts would be concluded and transitioned into a WMP for ongoing building system performance under the guidance of the healthcare organization's WMP team. Post occupancy, clinical surveillance [49] would be used to monitor the emergence of disease cases in complement to some periodic (monthly, quarterly, semi-annual, or annual) environmental testing.

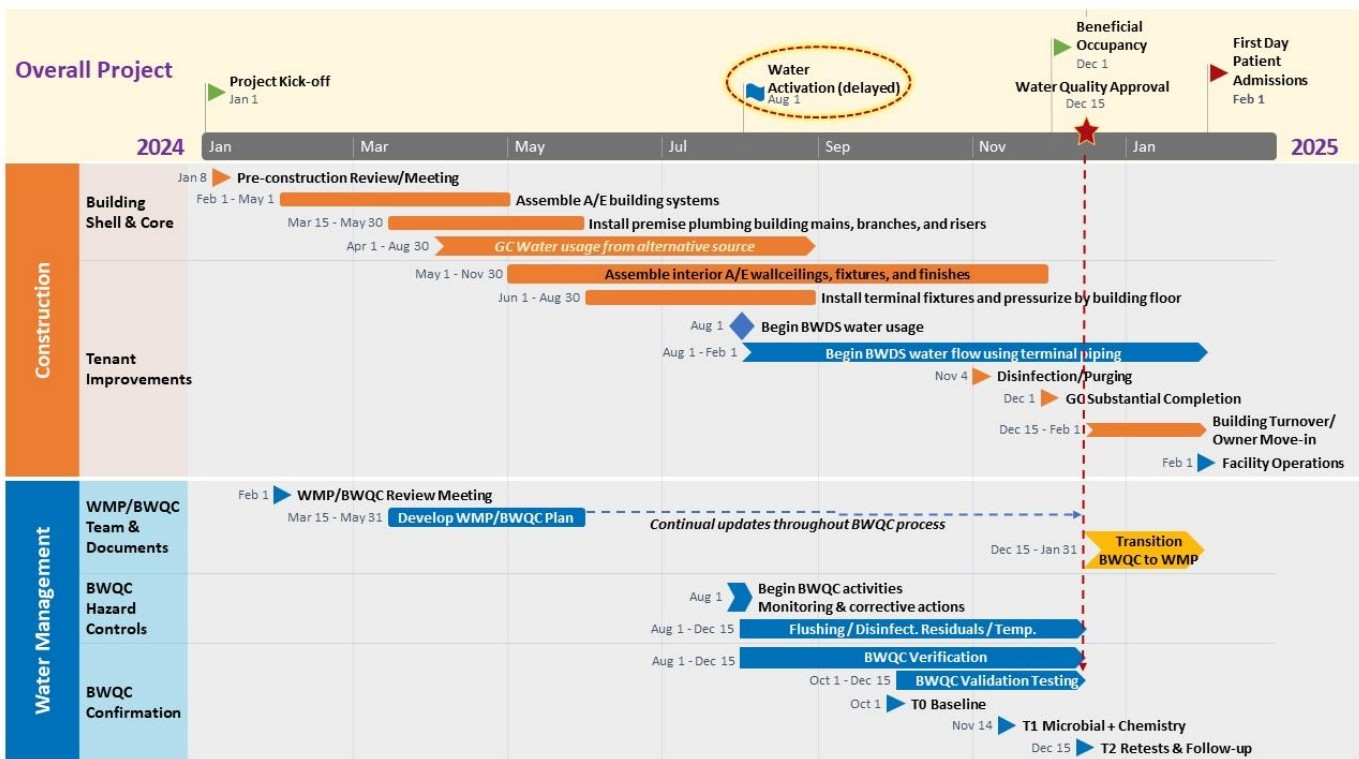

**Figure 5.** Gantt Schedule B—BWDS water activation is delayed until the later stage of the tenant improvement construction phase.

### 3.2. Adapting Gantt Chart for Project-Specific Results

The BWQC project schedule can be adapted and scaled to fit a wide variety of construction projects. First, the building owner would assemble the WMP team for the project and assign roles and responsibilities. Using Table 1 and its list of key tasks and milestones, the GC and BOR would develop a draft BWQC project schedule based on their project's specific construction timeline. The GC would review the Supplement S1 Building Water Quality Commissioning Project Schedule Checklist to determine if the construction project involves complex tasks beyond those listed in Table 1 (e.g., phased construction, utility shut-downs) and update the draft BWQC accordingly. The GC and BOR would then review the draft BWQC project schedule with the balance of the WMP team. The interdisciplinary WMP team members (e.g., A/E, CL, CxA, FM, IPC, OES, WMS) can provide expertise concerning other timeline and water management best practice details (e.g., patient safety, water management hazard controls, verification and validation, laboratory testing events) [6,21]. The CG should avoid using boilerplate documentation and scheduling techniques [35]. Boilerplate information typically results in copying irrelevant information from project to project without customization, which leads to misinformation, confusion, and schedule delays [35]. Although these methods were designed for complex healthcare environments, the same BWQC process can be adjusted at the WMP team's discretion for outpatient healthcare environments and even non-healthcare environments (e.g., schools, universities, hotels) where water quality and safety matters are also of concern.

### 4. Discussion

If A/E/C and CxA professionals are to be knowledgeable about BWDS waterborne pathogen prevention, then it seems methods and tools have to be developed using techniques familiar to the construction industry to avoid unintended consequences impacting patient safety. Implementing a BWQC project scheduling plan provides the team with a communication tool that is simple, clear, and adaptable to project-specific conditions. The BWQC project schedule contains relevant information to facilitate project manage-

ment and communication between the building owner's representatives and the A/E/C professionals. Assuming the BWQC is prepared and incorporated into the general contractor's construction management software programs, these activities can be connected to construction calendars, emails, requests for information, submittals, and other standard project documentation and communications [35]. Integrating digital information allows the construction team to assign additional labor resources, raise accountability, navigate delays, and achieve key milestones to ensure contractual obligations and project closeouts are completed [35].

Commissioning, including water quality and safety, is either encouraged or required in many countries for healthcare building systems to remove hidden threats and define unintended consequences prior to admitting patients [12,15,16,19,32,50]. However, after a review of ASHRAE Standards 188 and 514, we found that a uniform project scheduling process and specific steps expected for the safe completion of BWQC have been minimally defined and largely left to the discretion of the healthcare organization and their WMP team members. For an A/E/C industry that is unfamiliar with water management practices and untrained in water science [2,21,51], this represents a substantial knowledge gap. The National Health Service of England has provided Health Technical Memorandum 04-01 Part A [19] as a more robust document with detailed descriptions for BWDS installation, testing, and commissioning. Yet, when disseminating this information at a water hygiene conference, the agency identified substantial challenges with resistance and a lack of awareness of the importance of water safety in healthcare settings [3]. This has led to a 'black hole' of governance emerging as to who is responsible for BWDS performance and testing for compliance [3]. Even this more definitive document does not mention the project schedule as a potential tool for communication and accountability during construction.

Commissioning is viewed as a form of quality assurance and can be helpful for training facility maintenance and engineering staff who will ultimately operate the building. A BWQC project schedule developed prior to implementing a healthcare construction project will provide an opportunity for a more uniform approach and better interdisciplinary team communications. Gantt charts, which are graphically depicted, are widely understood for complex projects [35]. A BWQC project schedule will likely improve BWDS water quality and safety coordination activities. Over time, this type of WMP project planning effort, when combined with an ICRA, should reduce disease cases and deaths from construction activities in healthcare settings [2]. Taking such precautions could also reduce the financial burden associated with these events [52,53].

### 4.1. Construction Activities without Commissioning

Unmanaged BWDS construction without commissioning has exposed patients to water contaminants upon opening new or renovated healthcare facilities and has resulted in unnecessary disease cases and deaths [2]. Baker et al. [42] reported 116 disease cases and 26 deaths associated with a *Mycobacterium abscessus* outbreak immediately after the facility opened and continued over the next 29 months. BWDS challenges were attributed to a water conservation goal for green building design certification. Similarly, in 2009, an Alabama hospital constructed a nine-story new patient care tower, which was occupied in phases as various departments were completed [40]. Within one month of initiating patient care operations, Legionellosis disease cases (nine patients and one family visitor) emerged in the newly constructed hematology–oncology unit. An epidemiological investigation identified low residual disinfectant readings at patient care sinks and no flushing activities from the end of construction until the first day of patient care operations as likely culprits for environmental *Legionella* growth and spread. In another case study, Johnson and colleagues [43] were forced to manage a sporadic clonal outbreak of *Sphingomonas koreensis* from 2005 to 2016 after the construction of the National Institutes of Health (NIH) Clinical Center. The authors speculated that stagnant water lingered in premise plumbing from the construction phase, which likely allowed this rarely reported waterborne pathogen to entrench itself within the BWDS. After a cluster of six *Sphingomonas* patient infections

emerged in 2016, an epidemiology investigation was undertaken to look at patient-related cases dating back to 2005, the year of the initial building opening.

In the United Kingdom, Inkster and colleagues [54] reported an 8-month epidemiology investigation to resolve the source of mature biofilm developing in the BWDS within the first three years of building occupancy at Queen Elizabeth Royal Hospital for Children. In 2018, the investigation identified 23 confirmed pediatric disease cases in the hematology–oncology unit. The mature biofilm was tested and yielded the presence of 60 species of Gram-negative bacteria within the BWDS, with an emphasis on a rare waterborne pathogen of interest, *Cupriavidus pauculus*. The investigative team identified an extensive list of errors and omissions connected to poor construction and commissioning practices, such as (1) high water age due to early water activation followed by low or no usage during construction activities; (2) lack of establishing appropriate BWDS temperature ranges; (3) premise plumbing installation with dead legs; and (4) debris in water tanks and subsequent corrosion, among others. The authors recommended water commissioning practices to avoid premature growth and spread of complex biofilms in a newly constructed BWDS and assure safe water for patient care operations. These studies [40,42,43,54] demonstrated a clear lack of water safety coordination between the building owners and A/E/C and CxA professionals, which likely led to environmental amplification of these pathogens that ultimately led to unnecessary patient disease cases and deaths.

### 4.2. Construction Activities with Pseudo Commissioning Efforts

A limited number of research teams have reported implementing hazard controls (e.g., flushing, temperature monitoring, or measuring residual disinfectant) during healthcare construction projects [25,39,55,56]. These studies attempted a form of pseudo (i.e., self-styled) water commissioning efforts with a similar aim to reduce the risk of premature biofilm formation and avoid unnecessary patient exposure to waterborne pathogens after occupying a recently opened healthcare facility. These case studies represent varying levels of success, setbacks, or failures.

In Italy, De Giglio and colleagues [39] established water quality and safety at a new university hospital facility prior to opening for patient care operations. In April 2020, a water management process was initiated and implemented over a six-week period, including BWDS temperature system monitoring, disinfection, and flushing protocols after two years of construction. Ninety-one water samples were tested at three distinct periods of time (T0, T1, and T2) to determine water safety compliance for coliforms: *Escherichia coli (E. coli)*, Enterococci, *Pseudomonas (P.) aeruginosa*, and *Legionella (L.) pneumophila*. A comparison of water sample positivity rates from T0, T1, and T2 showed a progressive decline for coliforms, *P. aeruginosa*, and *L. pneumophila*. Only *L. pneumophila* declined at a positivity rate considered statistically significant (Fisher's F Test *p*-value = 0.07) from T0 to T2. *E. coli* and Enterococci were minimally detected across all three time periods. Based on these results, the study recommended that all new healthcare facilities initiate a WMP prior to building occupancy and conduct environmental water sampling to reduce the risk of disease cases emerging soon after the first day of patient care operations.

In France, Lecointe and colleagues [55] established protocols for environmental sampling at the completion of construction and before the first day of patient care operations. The criteria for opening the facility and transferring patients consisted of maintaining adequate hot temperatures (>122 °F, >50 °C) throughout the BWDS to all distal fixture points-of-use and undetectable *L. pneumophila* by culture with detection limits of $<10^3$ cfu/L. Water was activated and used for a limited number of construction activities until the end of the project. Continuous supplemental disinfection (0.5 mL/L of free chlorine) was performed for 10 months. Flushing (2 to 5 min per fixture) was performed through the end of the construction. However, water temperatures and other water parameters (e.g., residual disinfectant) were not verified. The study reported taking water samples prior to patient occupancy and detected *L. pneumophila* in reservoirs. It was determined that 17 heat exchangers and 18 hot water loops were not verified for circulating adequate

hot water temperatures. The BWDS was subsequently repaired and calibrated for proper hot water temperature ranges. Follow-up analytical testing confirmed *L. pneumophila* was non-detected ($<10^3$ cfu/L) in order to transfer patients to the new building environment.

If a BWQC construction schedule, as described in this technical note, had been properly specified, organized, and implemented for any of these healthcare construction projects, we postulate that the presence of waterborne pathogens in the BWDS would likely have been reduced. Some studies [39,55] recommended hazard controls and analytical testing to verify and validate safe water for patient care operations. Yet, without formal coordination and documentation, it is unlikely a building owner and their construction team can remain accountable and demonstrate the proper timing and sequence to complete water safety tasks and milestones. The BWQC project scheduling process was designed to address these issues by providing a uniform and flexible process for adapting to a project-specific situation. US and international WMPs are required to have communication, coordination, and documentation of all water system activities to meet WMP confirmation criteria [7,27,57]. Construction project scheduling is not formally mentioned in the current WMP standards of practice; however, this is a logical method to complement the required WMP commissioning process [6,7]. Without effective construction project documentation, a healthcare institution will have no records to defend its actions in the course of an internal or external audit or epidemiological investigation into disease cases or deaths from BWDS building occupant exposure.

### 4.3. A/E/C and CxA Professional Liability

Water and health science disciplines have expressed challenges with A/E/C industry engagement to address the serious consequences of poor water quality and patient safety [11,26,27]. Ficheux and colleagues [25] reported a facility-wide BWDS spread and growth of *P. aeruginosa* in a new healthcare facility about to open in France. The post-construction mitigation strategies involved extensive water hazard controls and required a substantial redesign of the BWDS (e.g., removal of dead legs; removal of the water softener; change in faucets and fixtures; replacing aerators with laminar flow devices; and removing mixing valves located over 10 feet (3 m) from the point-of-use). These BWDS changes to control a waterborne pathogen after construction was completed had significant facility cost implications. These types of A/E/C errors and omissions have increased professional liability exposure related to BWDS litigation, suggesting financial damages or even criminal charges should be considered from (1) patient injury, disease, or death; (2) facility loss of income from the suspension of patient care operations; (3) reputational harm; or (4) corporate fraud [1,58]. These litigation risks could be minimized by early engagement of A/E/C and CxA professionals in better BWQC practices, inclusive of construction project scheduling.

To address these professional liability gaps, we recommend future consideration be given to incorporating WMP practices and the development of a BWQC project schedule into a healthcare organization's construction policies and general practices. Although how to rewrite construction policies is beyond the scope of this technical note, this effort would introduce new terminology and legal language into A/E/C contracts. Major language edits would likely occur within the contract's general conditions, owner's general requirements, supplemental instructions to the general contractor, and specifications. More specifically, these edits would likely alter infection control mitigation measures, interim life safety measures, medical equipment installation, project submittals, test and inspection requirements, utility disruption procedures, and overall project scheduling requirements, among others. Agencies should then further standardize WMP practices for construction and commissioning to obtain a healthcare facility operational license and accreditation.

### 4.4. Future Smart Building Water Systems

Future considerations should be given to smart BWDS technology and software systems for assisting building owner verification and validation requirements of the BWQC

and WMP processes. To avoid BWDS systemic contamination [1,3], building owners should encourage the development of and consider the use of smart technologies to complement and advance their BWQC process [59]. During the construction and commissioning phases of work, intelligent construction recordkeeping systems [60], building automation controls [36], and A/E/C building information modeling (BIM) systems [61,62] could be integrated to monitor the BWQC process. These digital and web-based project management tools often help the GC track critical dependencies between tasks, monitor resource allocation, map out where BWDS installation changes occurred, manage construction delays, or facilitate financial claims. Similarly, for post-construction ongoing facility operations, smart building systems could be used to track BWDS performance through a centralized automation system using remote sensors and artificial intelligence [36]. Smart buildings have been used to reduce energy consumption, lower operating costs, improve air quality, improve thermal occupant comfort, and increase security and data collection. To the authors' knowledge, water quality and safety monitoring for implementing flushing protocols, measuring water temperature at distal locations, or verifying residual disinfectant levels have not been developed as part of these smart building system considerations.

*4.5. Limitations*

The BWQC construction schedule, tables, and definitions presented by the authors are only an example. This technical note has not been tested or validated within an observational research study to collect data and report results. Rather, the authors gathered information from known standards and policies and reported evidence by others (i.e., cited sources) into a BWQC project schedule for future implementation. Future case studies are under consideration to report the results of implementing this method by healthcare organizations and their program for construction projects. For these reasons, each healthcare WMP team and associated A/E/C and CxA professionals must evaluate this method in a project-specific context, as well as the organization's WMP construction policies. Construction scheduling methods will need to be modified and adjusted for any AHJ (local, state, or national).

The healthcare organization and its agents assume the sole risk and full responsibility for any construction scheduling method and the consequences of implementation in healthcare or other environments. The authors make no representations or warranties about the suitability, completeness, reliability, legality, accuracy, or appropriateness of the information provided to reduce the likelihood of waterborne pathogens in any BWDS or the disease cases, injuries, or deaths that may emerge from BWDS construction or commissioning activities.

## 5. Conclusions

Prior studies without a comprehensive BWQC practice have resulted in premature biofilm in premise plumbing, immediately exposing patients to unnecessary risk upon opening newly constructed or renovated healthcare facilities. In order to lower risk, we recommend a WMP BWQC plan and construction project schedule, as demonstrated in the technical note, be developed prior to the start of project construction activities. Once the contractor activates water flow within the BWDS for any reason (i.e., building construction utility, plumbing installation, or taking over an already active water system under renovation), the BWDS should be managed for water quality and safety. Other healthcare building utility systems are routinely modified, tested, and inspected prior to returning the system for patient care operations [14]. The lack of a WMP BWQC process has led to morbidity and mortality for the past 50+ years when performing construction activities in patient care settings [1,2,63]. If uniform BWQC methods were adopted and implemented, clinical teams could begin new or resume existing building operations safely without experiencing life-threatening water-related patient safety issues.

**Supplementary Materials:** The following supporting information can be downloaded at: https://www.mdpi.com/article/10.3390/buildings13102533/s1, Supplement File S1: Building Water Quality Commissioning (BWQC) Project Schedule Checklist.

**Author Contributions:** The following summarizes individual contributions: conceptualization, M.M.S.; methodology, M.M.S. and J.L.G.; formal analysis, M.M.S. and J.L.G.; writing—original draft preparation, M.M.S., J.L.G. and K.A.R.; writing—review and editing, M.M.S., J.L.G. and K.A.R.; visualization, M.M.S. and J.L.G.; project administration, M.M.S. All authors have read and agreed to the published version of the manuscript.

**Funding:** This research received no external funding.

**Data Availability Statement:** All data are presented within the article or the Supplementary Materials.

**Conflicts of Interest:** The authors declare no conflict of interest.

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
