# Peer review of "Building Water Quality Commissioning in Healthcare Settings: Reducing Legionella and Water Contaminants Utilizing a Construction Scheduling Method"

_buildings, doi:10.3390/buildings13102533_

Round 1
Reviewer 1 Report
The article is indeed intriguing and well designed, effectively explaining the problem through graphical analysis. While the English language used throughout the article is generally commendable, the authors are advised to carefully review the syntax and grammar to correct possible errors. In addition, it is advisable that authors adhere to the recommended referencing style for in-text citations, tables, and the bibliographic list to improve the overall quality and consistency of the manuscript.
Minor edit
Author Response
Reviewer #1
The article is indeed intriguing and well designed, effectively explaining the problem through graphical analysis. While the English language used throughout the article is generally commendable, the authors are advised to carefully review the syntax and grammar to correct possible errors. In addition, it is advisable that authors adhere to the recommended referencing style for in-text citations, tables, and the bibliographic list to improve the overall quality and consistency of the manuscript.
Authors’ Response: Thank you for your supportive comments. We have had an additional English editor (not an author) review the manuscript. We have modified for English editing throughout the manuscript. Examples can be found on lines: 177 – 183, 226-231, Table 1; 291, 370 – 371, 462, 465, 480- 482, 504, and 520. In response to other review comments substantial text has been added and also checked by an English editor.
Additionally, we are very familiar with the MDPI author guidelines and formatting requirements from prior submissions. We believe we have adhered to those author guidelines; however, each journal “desk” editor typically will provide us with any formatting issues that might be unique to this article or we have to deal with related to using EndNote citation software that cannot be repaired until all other editing is completed. We agree to review any other “desk” editorial items after the peer-review comments are completed including any adjustments for tables and figures for changes.
Reviewer 2 Report
The paper proposes a study on Building Water Quality Commissioning in Healthcare Settings.
As the authors themselves state, architectural, engineering, and construction (A/E/C) professionals and commissioning agents (CxA) have minimal knowledge about water science, water contaminants, or analytical laboratory water analysis. Therefore the most interesting and original aspect of the paper consists in the proposal of an interdisciplinary approach to the problem of water management in buildings.
It is a fundamental issue in the relationship between buildings and users' health.
The proposal to introduce water control and management actions among the tools to be implemented for smart buildings also appears very interesting and correct.
Finally I would like to suggest to the authors of integrae and better specify the description of the method used.
Author Response
Reviewer #2 Comment:
I would like to suggest to the authors to integrate and better specify the description of the method used.
Authors’ Response: Thank you for your supportive comments. We have added more description to the methods section of the article. Please see lines 166 – 173; 177 – 183; 198 – 225; and Table 1 updates for Key Team Leader(s) for additional description of methods.
Reviewer 3 Report
The manuscript is very well written. The novelty of the research work is highlighted in the document. Objectives and methodology is clear. Results represent a contribution to the current state-of-the-art. In my opinion, it can be acceptable in the current form. Thank you!
Minor editing of English language required
Author Response
Reviewer #3 Comment
The manuscript is very well written. The novelty of the research work is highlighted in the document. Objectives and methodology are clear. Results represent a contribution to the current state-of-the-art. In my opinion, it can be acceptable in the current form. Thank you!
Authors’ Response: Thank you for your supportive comments and suggesting the manuscript be published in its current state. Adjustments were made based on other reviewer comments that simply enhance the current direction of the manuscript.
Reviewer 4 Report
1. The introduction should provide more context and background information on the issue of waterborne pathogens in healthcare settings and the importance of addressing this risk.
2. The statement that the lack of a building water quality commissioning (BWQC) process is a substantial construction risk factor associated with disease cases and deaths should be supported by specific evidence or references.
3. The technical note on the BWQC schedule method should be described in more detail, including its purpose, methodology, and any validation or testing that has been conducted.
4. The benefits and advantages of implementing a BWQC schedule method should be clearly outlined, including how it helps healthcare organizations meet commissioning criteria and regulatory requirements.
5. The use of Office Timeline® to create a Gantt chart for the BWQC schedule should be explained further, including how it enhances project management and communication.
6. The roles and responsibilities of design and construction professionals, as well as healthcare organizations, in implementing the BWQC construction schedule method should be clarified.
7. Recommendations for customization of the BWQC schedule method for project-specific implementation should be provided, including any considerations or best practices for adapting it to different types of healthcare facilities.
8. The potential impact of incorporating the BWQC schedule method into an organization's construction policies should be discussed, including how it can contribute to standardized practices and improved water quality management.
Minor editing of English language required.
Author Response
Reviewer #4 Comments w/Author Responses
Authors’ Response: Thank you for all of your comments. We complied with the reviewer’s requests where appropriate and explained otherwise if we thought something may have been misunderstood or a change was not necessary.
- The introduction should provide more context and background information on the issue of waterborne pathogens in healthcare settings and the importance of addressing this risk.
Authors’ Response: Complied. See lines 31 through lines 45.
- The statement that the lack of a building water quality commissioning (BWQC) process is a substantial construction risk factor associated with disease cases and deaths should be supported by specific evidence or references.
Authors’ Response: Complied. See lines 31 through lines 45.
Additional note: articles were already referenced that contained such evidence. See citations #1 and #2. To comply further, we have articulated data from those studies and added other clarifying language and citations for introducing this topic to the A/E/C industry directly in the article.
- The technical note on the BWQC schedule method should be described in more detail, including its purpose, methodology, and any validation or testing that has been conducted.
Authors’ Response: Complied. See text inserted on various lines throughout the manuscript as follows:
Purpose: lines 111- 114; lines 166 – 173
Methodology: lines 166 – 173; lines 177 – 183; (see additional explanation under reviewer comment #5).
Any data related to validation or testing: listed additional language under the limitations section. See lines 519 – 526.
See additional responses below for other text that was added to the methods section related to reviewer comment #6.
- The benefits and advantages of implementing a BWQC schedule method should be clearly outlined, including how it helps healthcare organizations meet commissioning criteria and regulatory requirements.
Authors’ Response: updated the discussion section at the beginning for this language. See lines 337 – 363; lines 366 – 372.
- The use of Office Timeline® to create a Gantt chart for the BWQC schedule should be explained further, including how it enhances project management and communication.
Authors’ Response: Complied. We edited the Discussion Section lines for how scheduling enhances project management and communication. See lines 337 – 372; and text added lines 458 – 460.
Regarding Office Timeline ® - The reader can use any Gantt chart/bar chart project scheduling software program to achieve similar results. See lines 177- 183 for a better explanation of the authors’ intent as to the use of Gantt chart software. Office Timeline ® is a basic Gantt /bar chart program much like Microsoft Excel® is a basic spreadsheet program. Explaining in further detail how to use Office Timeline respectfully is not germane to obtaining a result. We simply declared the software we used as a minimal threshold of having some sort of digital method to build a graphic representation of a project schedule timeline.
- The roles and responsibilities of design and construction professionals, as well as healthcare organizations, in implementing the BWQC construction schedule method should be clarified.
Authors’ Response: Complied. See new section 2.3 BWQC Roles and Responsibilities lines 198 – 225. Also see updated Table 1 with new added column for suggested preliminary Key Team Leader(s) for each task/milestone.
- Recommendations for customization of the BWQC schedule method for project-specific implementation should be provided, including any considerations or best practices for adapting it to different types of healthcare facilities.
Authors’ Response: Complied. See new section and text description 3.2 Adapting Gantt Chart for Project Specific Results. See lines 312 – 332.
- The potential impact of incorporating the BWQC schedule method into an organization's construction policies should be discussed, including how it can contribute to standardized practices and improved water quality management.
Authors’ Response: Complied. See lines 485 – 497.
Round 2
Reviewer 4 Report
The new version is fine. It is suitable for publication.